

# Invasion genetics of the mummichog (*Fundulus heteroclitus*): recent anthropogenic introduction in Iberia

Teófilo Morim[1], Grant R. Bigg[2], Pedro M. Madeira[1], Jorge Palma[1], David D. Duvernell[3], Enric Gisbert[4], Regina L. Cunha[1] and Rita Castilho[1]

[1] Centre for Marine Sciences (CCMAR), University of Algarve, Faro, Portugal
[2] Department of Geography, University of Sheffield, Sheffield, United Kingdom
[3] Department of Biological Sciences, Missouri University of Science and Technology, Rolla, MO, United States of America
[4] IRTA, Aquaculture Program, Centre de Sant Carles de la Ràpita, Sant Carles de la Ràpita, Spain

## ABSTRACT

Human activities such as trade and transport have increased considerably in the last decades, greatly facilitating the introduction and spread of non-native species at a global level. In the Iberian Peninsula, *Fundulus heteroclitus*, a small euryhaline coastal fish with short dispersal, was found for the first time in the mid-1970s. Since then, *F. heteroclitus* has undergone range expansions, colonizing the southern region of Portugal, southwestern coast of Spain and the Ebro Delta in the Mediterranean Sea. Cytochrome *b* sequences were used to elucidate the species invasion pathway in Iberia. Three Iberian locations (Faro, Cádiz and Ebro Delta) and 13 other locations along the native range of *F. heteroclitus* in North America were sampled. Results revealed a single haplotype, common to all invasive populations, which can be traced to the northern region of the species' native range. We posit that the origin of the founder individuals is between New York and Nova Scotia. Additionally, the lack of genetic structure within Iberia is consistent with a recent invasion scenario and a strong founder effect. We suggest the most probable introduction vector is associated with the aquarium trade. We further discuss the hypothesis of a second human-mediated introduction responsible for the establishment of individuals in the Ebro Delta supported by the absence of adequate muddy habitats linking Cádiz and the Ebro Delta. Although the species has a high tolerance to salinity and temperature, ecological niche modelling indicates that benthic habitat constraints prevent along-shore colonisation suggesting that such expansions would need to be aided by human release.

## INTRODUCTION

As a consequence of human activities involving large distance marine transportation and trade, worldwide marine biological invasion rates have increased dramatically in the last 30 years (e.g., *Hulme, 2009*). Along the European coasts, there are reports of over 850 invasive species, 237 of which occur along the eastern Atlantic coastal areas, and 680 in the Mediterranean Sea and the remaining in the Baltic Sea (*Galil et al., 2014*). Marine invasive

Corresponding author
Rita Castilho, rita.castil@gmail.com

species pose a significant environmental threat as they are recognized as one of the major drivers of biodiversity loss (*Millennium Ecosystem Assessment, 2005*), altering ecosystems and their dynamics, shifting the community structure and displacing endemic species (*Molnar et al., 2008*). Negative impacts may also be registered at the economic and social levels, affecting fisheries, aquaculture, tourism or human health (*Bax et al., 2003*). Invasive species spread and occupy new marine and coastal ecosystems through several maritime introduction vectors such as ballast water, biofouling of vessels, aquaculture escape or ornamental species trade (see *Williams et al., 2013* and references therein).

Reconstructing the invasion pathways and identifying the putative source populations with historical and contemporary vector records is a difficult task (*Estoup & Guillemaud, 2010*; *Lawson Handley et al., 2011*). In this context, molecular genetic data is a powerful tool to reconstruct invasive history by identifying putative source populations and genetic bottlenecks (*Bock et al., 2015*; *Cristescu, 2015*). Although genetic data does not always allow for successful identification of these processes, there are three conditions which increase the probability of accurate reconstruction of the invasive pathway: (1) extensive sampling along the native range to ensure all the putative source populations are known, (2) the presence of genetic structure in the native range to narrow down the putative source regions and (3) a short amount of time passed since the invasion so that processes such as genetic drift do not increase the genetic differentiation between native and invasive populations (*Geller, Darling & Carlton, 2010*).

The advent of ecological niche modelling (ENM) has contributed to the building of environmental risk maps for biological invasion by allowing formal and quantitative inclusion of varying environmental characteristics, both in space and time, to be added to such considerations. Such new approaches to invasion risk monitoring provide useful insights by predicting both potential colonization routes and the probability of new invasions occurring (*Hulme, 2009*; *Molnar et al., 2008*).

The mummichog, *Fundulus heteroclitus* (Linnaeus, 1766), is a small teleost fish naturally occurring almost continuously in saltmarshes of the North American east coast, from Newfoundland to Florida (*Hardy Jr, 1978*). This species is extremely resistant to a wide range of salinities and temperatures, and can be found in freshwater, brackish or saltwater, inhabiting sheltered coastal areas such as saltmarshes, tidal creeks, estuaries or bays all year-round (*Bigelow & Schroeder, 1953*; *Hardy Jr, 1978*). *F. heteroclitus* is one of the most stationary marine species (*Bigelow & Schroeder, 1953*), with short dispersal distances (1–2 km, *Fritz, Meredith & Lotrich, 1975*), high site fidelity closely related to the presence of saltmarshes (*Kneib, 1984*) and short home ranges (36–38 m, *Lotrich, 1975*). Enzyme-coding loci (*Powers & Place, 1978*; *Powers et al., 1986*; *Ropson, Brown & Powers, 1990*), mitochondrial DNA (*Bernardi, Sordino & Powers, 1993*; *González-Vilaseñor & Powers, 1990*; *Smith, Chapman & Powers, 1998*), putative neutral nuclear microsatellite loci (*Adams, Lindmeier & Duvernell, 2006*; *Duvernell et al., 2008*), and a variety of nuclear single-nucleotide polymorphisms (see *McKenzie, Dhillon & Schulte, 2016* and references therein) demonstrated concordance of clinal patterns, where a break is placed between the meridians 40° and 41°N, along the coast of New Jersey. These studies concur with *Morin & Able (1983)*, supporting the division into two subspecies: *F. heteroclitus macrolepidotus*

(Walbaum, 1792) to the north of the cline and *F. heteroclitus heteroclitus* (Linnaeus, 1766) to the south of the cline.

In the Iberian Peninsula, *F. heteroclitus* was first detected in the 1970s, in the Guadalquivir and Guadiana saltmarshes (southwestern coast of Spain) (*Hernando, 1975*) and in the Guadiana Delta (*Coelho, Gomes & Ré, 1976*). More recently, its presence was also reported in the Ebro Delta, northeastern coast of Spain (*Gisbert & López, 2007*) and in the Ria Formosa lagoon, south of Portugal (e.g., *Catry et al., 2006*) where it can reach high densities (*Gonçalves et al., 2017*). The species was probably introduced between 1970 and 1973 in the Spanish saltmarshes (*Fernández-Delgado, 1989*) either involuntarily via aquarium trade (e.g., *Elvira & Almodóvar, 2001*), ballast water (*Fernández-Delgado, 2010*; *García-Revillo & Fernández-Delgado, 2009*), or intentionally for purposes of biological control (*Gozlan, 2010*). The Ebro Delta individuals were probably caught in the southern Spanish saltmarshes to be used in aquaculture and the aquarium trade (*Gisbert & López, 2007*) and were later released from captivity. Another possibility for the Ebro Delta introduction is an accidental escape from a research centre nearby, where this species had already been used as a scientific model (*Gisbert & López, 2007*). Two previous studies have analysed the origin of the southwestern Spanish populations using mitochondrial DNA (mtDNA), concluding the founder individuals were original from the northern native region between Maine and Nova Scotia where the northern subspecies *F. h. macrolepidotus* is distributed (*Bernardi et al., 1995*; *Fernández-Pedrosa, Latorre & González, 1996*).

In the present study, we aim to build on previously published studies on the invasive range of *F. heteroclitus* by using (1) more sampling locations (one in the Mediterranean Sea and two locations in the eastern Atlantic, and 13 native locations), (2) a significantly larger number of individuals (248 in total), and (3) a three times larger fragment of the mitochondrial DNA cytochrome *b* gene; and to evaluate the genetic diversity and invasion pathways. Given that the species was recently reported in the Iberian Peninsula and has a limited adult dispersal capability, we tested the hypothesis of a human-mediated single Iberian introduction followed by dispersal promoted along the main oceanographic currents. and we mapped the environments compatible with the species' ecological requirements to evaluate the dispersion potential through suitable continuous habitat. This hypothesis leads to the expectation of an Iberian invasion based on a few founder individuals, with consequent lower genetic diversity than the putative identified source population. Also, the Iberian populations are expected to show no evident genetic structure given the short time since invasion.

## MATERIAL AND METHODS

### Sampling

A total of 248 *Fundulus heteroclitus* individuals from 16 locations: 13 sites in the western Atlantic, one in the Mediterranean Sea and two locations in the eastern Atlantic (Table 1 and Figs. 1A, 1B) were obtained and stored in 96% ethanol and kept at −20 °C. The populations from the western Atlantic sampled above 40°N are hereafter referred to as northern locations (ID 1-8), while samples collected below that latitude are referred to

**Table 1** Sample location, sample abbreviations and summary statistics for a cytochrome *b* sequence fragment from *Fundulus heteroclitus*. ID refers to numbers in Fig. 1.

| Location | ID | Code | Latitude/longitude | $n$ | $n_h$ | $n_p$ |
|---|---|---|---|---|---|---|
| Bridgewater | 1 | HV | 44°22.0′N/64°31.0′W | 15 | 2 | 0 |
| Chewonki | 2 | CM | 43°57.3′N/69°43.2′W | 15 | 3 | 1 |
| Wells | 3 | WM | 43°19.2′N/70°34.2′W | 15 | 4 | 2 |
| Woods Hole | 4 | WH | 41°31.5′N/70°40.4′W | 16 | 9 | 4 |
| Jerusalem | 5 | JR | 41°23.1′N/71°31.5′W | 15 | 5 | 4 |
| Clinton | 6 | CC | 41°15.3′N/72°32.8′W | 16 | 7 | 6 |
| Newark Bay | 7 | NB | 40°41.2′N/74°06.7′W | 15 | 8 | 6 |
| Red Bank | 8 | RE | 40°20.9′N/74°05.0′W | 15 | 7 | 5 |
| Tuckerton | 9 | TN | 39°32.2′N/74°19.4′W | 15 | 10 | 8 |
| Speace | 10 | SP | 38°09.1′N/75°17.2′W | 15 | 9 | 6 |
| Suffolk | 11 | CH | 36°51.8′N/76°28.7′W | 16 | 7 | 3 |
| Roanoke Island | 12 | RI | 35°53.8′N/75°36.9′W | 15 | 9 | 7 |
| Skidaway Island | 13 | SI | 31°56.8′N/81°04.2′W | 16 | 11 | 1 |
| Faro | 14 | RF | 37°00.3′N/07°58.0′W | 16 | 1 | 0 |
| Cádiz | 15 | CD | 36°31.4′N/06°11.4′W | 17 | 1 | 0 |
| Ebro Delta | 16 | ED | 40°37.38′N/0°39.44′E | 16 | 1 | 0 |

**Notes.**

$n$, number of individuals; $n_h$, number of haplotypes; $n_p$, number of private haplotypes.

as southern locations (ID 9-13). Samples collected from the Mediterranean and eastern Atlantic are referred to as Iberian (ID 14-16).

## DNA extraction, PCR amplification and sequencing

Total genomic DNA was extracted from caudal fin tissue following a standard Chelex 100 protocol (*Walsh, Metzger & Higuchi, 1991*). Extraction results were checked by electrophoresis in 0.8% agarose gel stained with GelRed. Polymerase Chain Reactions (PCR) were conducted in a total volume of 25 µL, with 1X buffer, 10 mM dNTPs, 10mM of each primer, 1U Taq Advantage 2 Polymerase mix DNA polymerase (CLONTECH-TaKaRa), 2 µL of DNA and Milli-Q water to the final volume. A fragment of the cytochrome *b* (cyt *b*) gene (1,000 base pairs) was amplified with the forward primer GludG-L14724 (*Palumbi et al., 1991*) and the reverse primer cb6b.h (*Martin & Bermingham, 1998*). PCR amplification consists of an initial 4 min denaturation step at 95 °C, followed by 40 cycles of 1 min at 94 °C (denaturation), 1 min at 50 °C (annealing) and 1.5 min at 72 °C (extension), and a 5 min final extension step. When amplification was not successful the following profile was used: initial 3 min denaturation step at 95 °C, followed by 32 cycles of denaturation for 30 s at 95 °C, annealing for 30 s at 54 °C and extension for 1 min at 68 °C, and a final extension step for 4 min at 68 °C. PCR products were checked afterwards by electrophoresis in a 1% agarose gel stained with GelRed. Mitochondrial DNA was purified by ethanol/sodium acetate precipitation (*Sambrook & Russel, 2001*). Its purity and quantity were analysed using a NanoDrop1000 spectrophotometer (Thermo Fisher Scientific, Waltham, MA, USA). Sequencing was performed on an ABI 3130xl capillary

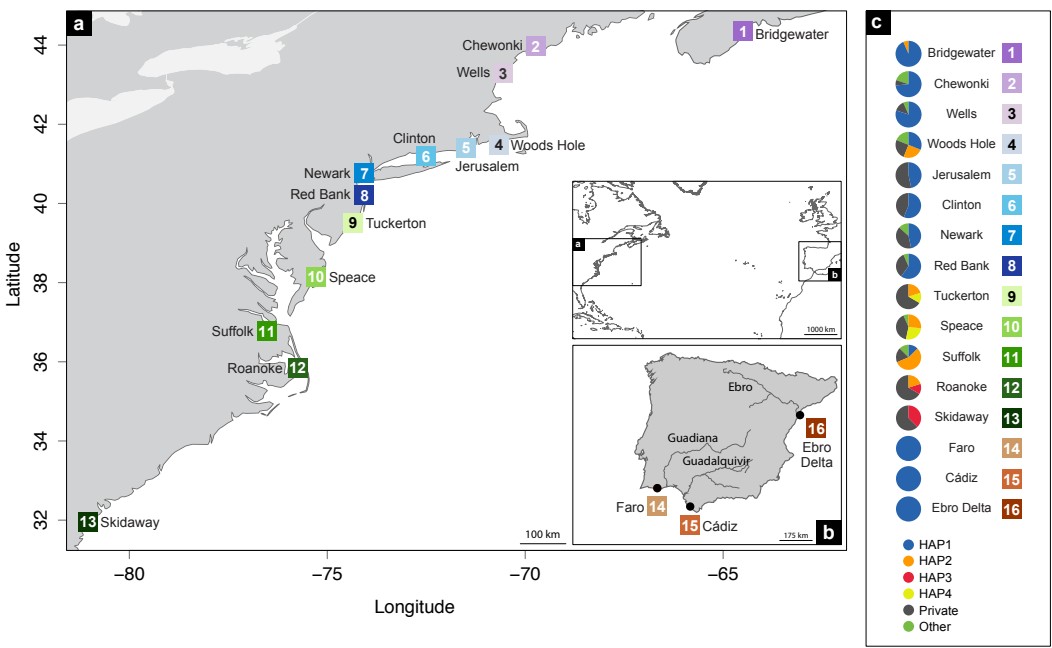

**Figure 1** **Distribution map of localities and haplotypes.** Distribution map of localities along the coast of (A) North America, and (B) Iberian Peninsula of *Fundulus heteroclitus*. Locations and sample details can be found in Table 1. (C) Coloured circles display the distribution of the relative proportions of the cytochrome *b* haplotypes from each location: the four most common haplotypes, the private haplotypes, and all the other shared, but less frequent haplotypes.

sequencer (Applied Biosystems –CCMAR, Portugal) using the forward primer from the PCR amplification (GludG-L14724).

## Genetic analysis

Cyt *b* sequences were aligned and manually checked using the software Geneious v4.8.2 (Biomatters, Ltd., Auckland, New Zealand). The number of haplotypes (*n*), number of private haplotypes (*n_p*), and the haplotype (*h*) (*Nei & Tajima, 1981*) and nucleotide diversities ($\pi$) (*Nei, 1987*) were calculated for each location using the DnaSP v5.10.1 (*Librado & Rozas, 2009*). All sequences were uploaded to GenBank (MH809691–MH809938). To represent the phylogeographic relationships among haplotypes, a haplotype network was constructed using the Median Joining algorithm implemented in NETWORK v5.0 (*Bandelt, Forster & Röhl, 1999*, fluxus-engineering.com).

We used two approaches to infer the most probable source area of the introduced Iberian populations within the native range of *F. heteroclitus*: (1) the geographical distribution of haplotypes in native populations, and (2) the phylogeographical relationships among haplotypes.

## Ecological niche modelling

To explore the potential spread of *F. heteroclitus* within European waters beyond its currently known locations we used an ecological niche model. We selected the interactive AquaMaps system in FISHBASE (http://www.fishbase.org). Using observed locations of

**Table 2  Aquamaps default environmental envelope for *F. heteroclitus*.**

| Variable | Absolute minimum | Preferred minimum (10th percentile) | Preferred maximum (90th percentile) | Absolute maximum |
|---|---|---|---|---|
| Depth [m] | 0 | 0 | 3 | 5 |
| Sea Surface Temperature [SST; °C] | 4.76 | 6.41 | 23.53 | 26.1 |
| Sea Surface Salinity [SSS; psu] | 28.96 | 29.94 | 35.74 | 39.6 |

adults to construct a range of acceptable environmental conditions within which a species can exist, this model uses a c-squares distribution modelling approach (*Rees, 2003*) to predict the probability of occurrence of the adult fish for a particular location, possessing specific environmental parameters. While this is normally computed just within the native range of a species it can be extended within the interactive system to consider all possible locations, and the associated probability of occurrence linked to the location's mean environmental state if the species was able to reach that region. The basic statistical approach is given in *Kaschner et al. (2006)* and the AquaMaps version is described in *Kesner-Reyes et al. (2012)*. Another ecological niche model, based on maximum entropy principles (MAXENT, *Phillips & Dudík, 2008*), was also tested using environmental parameters of sea surface temperature (SST) and sea surface salinity (SSS), but the specialist ecological substrate niche of *F. heteroclitus* led to poor solutions for range prediction using ENM. This latter approach is therefore not considered further here.

A set of favourable and extreme environmental conditions compatible with the native occurrence of *F. heteroclitus* is automatically specified by AquaMaps, relying heavily on *Page & Burr (2011)*. It includes parameter ranges for water depth, SST, SSS, primary production rates and sea-ice cover, leading to probabilities of occurrence exceeding 0.6 along the whole eastern seaboard of North America from South Carolina to the Canadian Maritime Provinces south of the Gulf of St. Lawrence. This matches well to the known range (*Page & Burr, 2011*). However, the long-term survival of *F. heteroclitus* in two Iberian estuaries has implications for extending the extreme salinity level that this species can tolerate beyond that automatically specified. The extreme salinity tolerance has therefore been raised to 39.6 psu, compared to the automatic 36.47 psu initially prescribed by AquaMaps, consistent with the values found for the Ria Formosa (*Cristina et al., 2016*). In addition, the sea-ice variable has been excluded from constraining the AquaMaps solution, as this is not relevant to the area being considered in this study, and also the primary production variable, as coastal environments satisfy the automatic requirement almost uniformly. The environmental variables used for the AquaMaps simulation shown here are given in Table 2.

AquaMaps was re-run with these environmental constraints and using the combined occurrence data from the native range, along the North American eastern seaboard, and the two Iberian sites of Ria Formosa and the Ebro River Delta. There is no change to the North American predictions, so here we concentrate only on those for European waters. The occurrence of *F. heteroclitus* was further constrained to coastal zones with the muddy benthic habitats in which *F. heteroclitus* is found. These are defined as the coastal fine

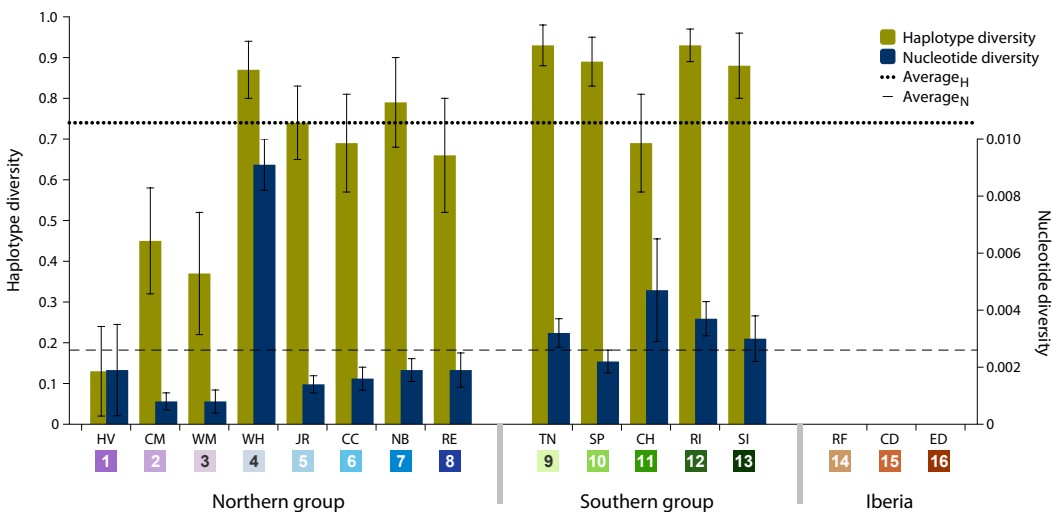

**Figure 2 Haplotype and nucleotide diversity.** Haplotype and nucleotide diversity of *Fundulus heteroclitus* from all sampled locations. Abbreviations in the legend are defined in Table 1, and colours are the same as in Fig. 1.

muds to muddy sands benthic environments of the EMODnet Seabed Habitats project (http://www.emodnet-seabedhabitats.eu). These environments are formally defined by *Long (2006)*, but basically include sediments with more than a 10% mud component.

## RESULTS

Data analysis on the sampled specimens resulted in cyt *b* sequences with a final length of 700 base pairs (bp), which comprised 77 (11%) polymorphic sites and 32 (41.6%) parsimony-informative sites. These polymorphisms defined 70 haplotypes, of which 62 (88.6%) are private haplotypes (present in one location only) and 55 (78.6%) are singletons (present in one individual only). Overall haplotype diversity was high (0.74 ± SD 0.03), ranging from null in Iberia (ID 14-16) to 0.93 in Tuckerton (ID 9) and Roanoke Island (ID 12), whereas mean nucleotide diversity was low (0.26% ± SD 0.06%) ranging from null diversity in Iberia to 0.91% in Woods Hole (ID 4) (Table 1 and Fig. 2).

The most abundant haplotype in North American locations is shared by 50.4% ($N = 125$) of the individuals and is present in all northern group locations, in one southern location and in all Iberian locations (Fig. 1C). This is the only haplotype detected in the invasive range of the species (Faro, Cádiz and Ebro Delta). The second most frequent haplotype in North American locations is shared by 9.7% ($N = 24$) individuals in six locations, although in higher frequency in the southern group locations. Two other haplotypes were found in 3.2% ($N = 8$) and in 2.4% ($N = 6$) of the individuals from two locations. All other haplotypes were present in five or less individuals and in less than three locations (Fig. 1C).

The cyt *b*- based haplotype network (Fig. 3) displays two haplogroups separated by nine mutational steps. Haplogroup A is constituted by all eight northern group locations (Bridgewater to Red Bank, ID 1-8), two individuals from the southern group (Suffolk, ID
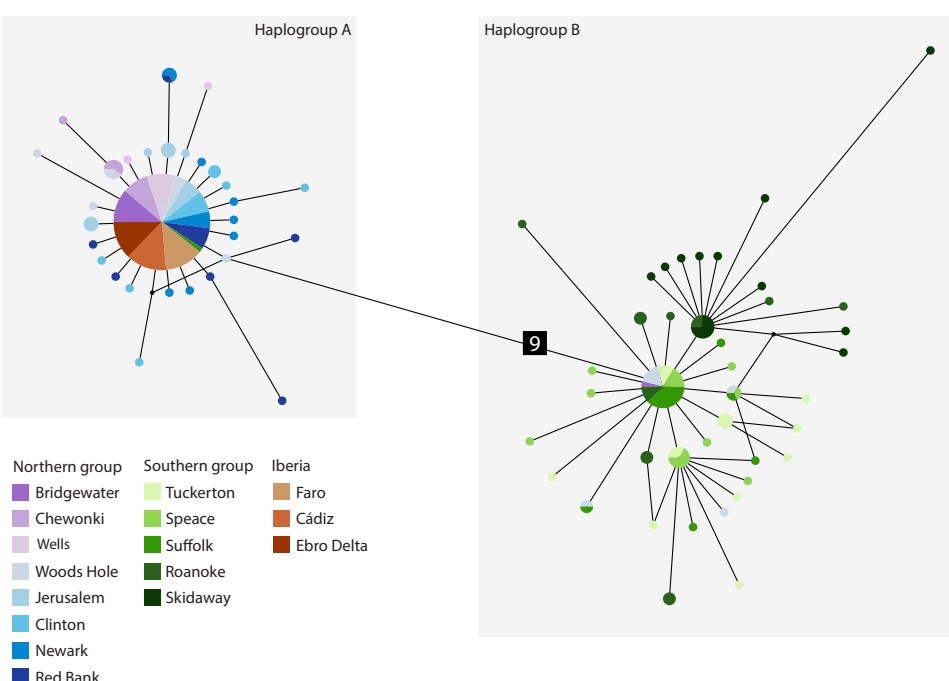

**Figure 3  Median-joining cytochrome *b* haplotype network for *Fundulus heroclitus*.** Median-joining cytochrome *b* haplotype network for *Fundulus heroclitus*. Each circle represents a different haplotype with size proportional to the frequency of the haplotype within the sample. Line length is proportional to the number of mutations between haplotypes. Each colour corresponds to a different location. Colours are the same as in Fig. 1. Locations details can be found in Table 1. The two black dots represent putative unsampled haplotypes, and the number in black square shows the number of mutations separating haplogroup A from the haplogroup B.

11) and by the Iberian locations (Faro to Ebro Delta, ID 14-16). Haplogroup B is formed by all the other individuals from all southern locations (Tuckerton to Skidaway Island, ID 9-13) and includes eight individuals from northern locations: one from Bridgewater (ID 1) and seven from Woods Hole (ID 4). Overall, both haplogroups display star-like configurations with different levels of complexity. The northern haplogroup (A) is simpler with 90% of the haplotypes separated by a single mutation, while the southern haplogroup (B) is more complex, with three mini-stars interconnected by one mutation each, and haplotypes separated by up to five mutations. The haplotype found in the Iberian Peninsula belongs to the northern haplogroup.

The AquaMaps ENM shows that the basic environment for the spread of *F. heteroclitus* is fundamentally favourable (probability > 0.75), along much of the Atlantic coastline of western Europe (Fig. 4). Conditions become less favourable in the Mediterranean, although the Alboran Sea, east of the Strait of Gibraltar, has an environment that is at least acceptable (probability > 0.25). There are also a small number of estuaries along the Balearic Sea coastline of NE Spain and southern France where acceptable conditions are also found. The main constraint on the spread of *F. heteroclitus*, however, is the absence of muddy benthic habitats. The latter are shown by the solid lines in Fig. 4. All the European

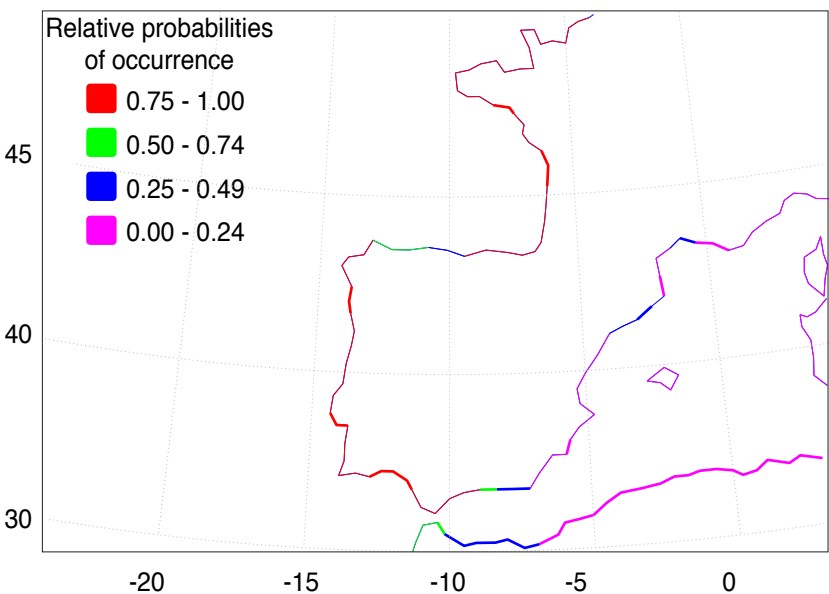

**Figure 4 Western European and Mediterranean coastal environments showing AquaMap probabilities of occurrence of *F. heteroclitus*.** Western European and Mediterranean coastal environments showing AquaMap probabilities of occurrence of *F. heteroclitus*. Those areas in bold show coastal seabed habitats with a mud content >10%.

coastlines within the zone where *F. heteroclitus* is now found have a small number of muddy estuaries separated by long stretches of unsuitable, rockier, benthic habitats. It is only along the environmentally unfavourable North African coast that extensive areas of favourable habitats are found. The ENM shows conclusively that lateral spread of *F. heteroclitus* along European shorelines by natural means is very unlikely.

## DISCUSSION

Results revealed the presence of a single haplotype common to all individuals in the Iberian Peninsula. This haplotype is the most abundant in the northern group of the native distribution, indicating the most probable origin of the invasion. We posit the most likely introduction vector to be the aquarium trade, and propose, through a combination of DNA and ENM evidence, that the Ebro Delta colonization results from an independent human-mediated secondary introduction. Before addressing the main interpretations and conclusions of these results, one main caveat must be addressed. Successful understanding of the invasion pathway relies on (1) comparable genetic data retrieved from an adequate number of sampled individuals throughout the entire native range, (2) presence of genetic clines within the native populations and (3) the use of adequate molecular markers (*Geller, Darling & Carlton, 2010*). While the first two points are fulfilled for *F. heteroclitus*, the use of a single mitochondrial DNA marker constitutes the main caveat of this study. Mitochondrial DNA has been a widely used molecular marker in population genetics studies (*Ballard & Whitlock, 2004*). Nevertheless, the use of high variable nuclear markers, such as microsatellites, provides an opportunity to perform assignment tests based on

their multiple-locus genotypes, to test for recent reductions in population sizes and to estimate effective population sizes. However, previously published studies show relatively low microsatellite genetic variation in the northern group (*Duvernell et al., 2008*). Single nucleotide polymorphisms (SNPs, *Morin et al., 2004*), extend the previously referred analytical possibilities improving their statistical power because of the sheer number of existing loci genome wide.

## Genetic diversity

The presence of a single haplotype common to all *F. heroclitus* sampled in the Iberian Peninsula lends support to the hypothesis of an extremely recent introduction of the species which has not allowed the accumulation of mutations at the mtDNA level, and with a single introduction event composed by a very small number of individuals (*Roman & Darling, 2007*). Theoretically, invasive species are expected to suffer loss of genetic variation since the new established populations are often based on a few individuals, which by definition, have lower genetic diversity than the native source populations (*Dlugosch & Parker, 2008*). The single-haplotype characteristic can be found in other invasive species, such as *Equulites elongatus,* the slender pony fish (*Sakinan, Karahan & Ok, 2017*); *Cercopagis pengoi*, the fishhook waterflea, a planktonic cladoceran crustacean (*Cristescu et al., 2001*); *Corbicula fluminea*, the Asian clam (*Gomes et al., 2016*) and *Didemnum perlucidum,* a sea squirt (*Dias et al., 2016*). However, many successful invasive species do not display significant erosion of genetic diversity (*Dlugosch & Parker, 2008*). For example, in a recent review of the literature on European sea invasion genetics, in 54% of studies that compared the genetic diversity between introduced species and their native range, 74% reported comparable levels of diversity between them, while only 23% displayed a reduction in the genetic diversity of introduced species, and the remaining 2% showed an increase in diversity (*Rius et al., 2014*).

## Population sources

We identified a single Iberian haplotype present in all northern populations in high frequency (between 47 and 93%) and in two individuals in Suffolk, one of the southern locations. According to our results, the northern group is the most probable source of the founder individuals, which corresponds to the natural range of the subspecies *F. h. macrolepidotus*. However, we cannot definitely exclude Suffolk as a presumptive population source. The absence of genetic diversity in the Iberian Peninsula populations prevents the precise determination of the putative source population. The low spatial resolution of our data arises from the lack of genetic variability in the invasive range, rather than insufficient sampling of *F. heroclitus* individuals in the Iberian Peninsula or in its native range (*Muirhead et al., 2008*).

Nevertheless, our findings are consistent with two previous studies on the origin of invasive individuals found in the Guadalquivir. First, based on mtDNA restriction fragment length polymorphisms (RFLP) (*Fernández-Pedrosa, Latorre & González, 1996*) reported the presence of two haplotypes: the most abundant corresponding to the northern haplotype 1, dominant between Maine and Nova Scotia in North America; the other haplotype did

not match any of the sampled native haplotypes and we found no evidence of its presence in the present study, using a larger number of individuals. It was previously suggested to be either a native unsampled haplotype or an endemic haplotype from Iberia, which is rather unlikely due to its recent invasion (*Fernández-Pedrosa, Latorre & González, 1996*) and total absence of records in the area. Although there are no reports of hybridization between *F. heteroclitus* and any of the Spanish endemic species, the presence of a new haplotype could nonetheless be due to hybridization (*Rius et al., 2014*). Secondly, a study based on cyt *b* sequences (*Bernardi et al., 1995*) concluded the individuals from the Guadalquivir originated in the region between Maine and Nova Scotia. While we cannot discount the possibility of a few rogues from Suffolk being responsible for the invasion, the weight of probability falls heavily on the side of Maine to Nova Scotia origin.

## Introduction vector

Since it was first recorded in Iberian saltmarshes, several studies linked *F. heteroclitus* introduction to different vectors. The aquarium trade has been suggested as the most important vector responsible for the introduction of this species (e.g., *Gozlan, 2010* and references therein), followed by ballast water (*Fernández-Delgado, 2010*; *García-Revillo & Fernández-Delgado, 2009*), biological control (*Gozlan, 2010*) and unknown origins (*Fernández-Delgado, 1989*; *Gutiérrez-Estrada et al., 1998*). We posit that the introduction via the aquarium trade or animal acquisition for scientific purposes is the most probable scenario responsible for the establishment of the first individuals in Iberia, given the genetic results obtained.

It is well known that vessels can transport large numbers of organisms from several species at the same time in their ballast water (*Gollasch, 2007*). Since there are countless vessels active around the world (e.g., *Kaluza et al., 2010*), ballast water-mediated transport allows the possibility of multiple introduction events, each with large groups of individuals (*Hulme, 2009*). Considering such large groups usually comprise higher genetic diversity than fewer individuals alone, the assemblages transported are likely to display levels of genetic diversity similar to the levels found within their native range (*Wilson et al., 2009*). Our findings, however, suggest that this was not the introduction vector responsible for the spread of the species; in contrast to the expectation of similar levels of genetic diversity between the native and invasive range, the Iberian locations display a strong founder effect, with all samples sharing a single haplotype.

Similarly to ballast water, the aquarium and ornamentals trade transport many species at a global scale (*Padilla & Williams, 2004*). For instance, at least 19% of the invasive fishes found in the Iberian Peninsula were introduced via the aquarium trade (*Maceda-Veiga et al., 2013*). However, there are two main differences between introductions that follow ballast water or the aquarium trade. First, each introduction event after aquarium release is likely to comprise a small number of individuals (*Duggan, Rixon & MacIsaac, 2006*). Thus, even though this vector may be responsible for the establishment of several non-native species (*Padilla & Williams, 2004*), multiple introductions would be necessary for an invasive species to display high genetic diversity (*Roman & Darling, 2007*). Secondly, the individuals released by aquarists are usually adults of higher fitness, which makes them

better adapted to survive in a natural environment (*Padilla & Williams, 2004*). Thus, not only are *F. heteroclitus*'s invasive genetic diversity and structure consistent with an introduction of a low number of individuals via the aquarium trade, it is also plausible that a few resistant individuals would manage to survive, reproduce and colonize the environment in which they were released.

## Human mediated introduction in the Ebro Delta

Although the absence of genetic structure within Iberia limits possible insights into the invasion pathway, our data support the hypothesis of a human-mediated introduction episode responsible for the establishment of *F. heteroclitus* in the Ebro Delta, as previously suggested by *Gisbert & López (2007)* based on taxonomic identification. The hypothesis of a long-distance colonization via natural dispersal is quite unlikely, as is strongly shown by the ENM analysis (Fig. 4).

According to a review of the geographical distribution of Cyprinodontiformes along the northeastern coast of Spain by *García-Berthou & Moreno-Amich (1991)*, no *F. heteroclitus* individuals were found at the Ebro Delta in 1989; the first record of the species was only registered 16 years later by *Gisbert & López (2007)*. Thus, we estimate the date of establishment in the Ebro Delta ranges between 12 and 28 years ago. Assuming this estimate is correct, a natural colonization hypothesis implies individuals would have taken roughly two decades to travel more than 1,000 km from their southernmost limit located in the Guadalquivir saltmarshes (*Gutiérrez-Estrada et al., 1998*) to the Ebro Delta. However, when compared with the natural colonization of the Ria Formosa, this hypothesis seems quite improbable. While *F. heteroclitus* was never collected during sampling events that happened in the Ria Formosa between 1980 and 2006 (*França, Costa & Cabral, 2009*; *Ribeiro et al., 2006*; *Ribeiro et al., 2008*), analysis of prey remains left by Little Terns (*Sterna albifrons*) in the salt-pans and barrier islands revealed this prey species was present in the salt-pans and adjacent channels at least since 2002 (*Catry et al., 2006*). Although this may sound contradictory, *F. heteroclitus* could in fact have been present in the Ria Formosa in specific unsampled locations or at extremely low densities, avoiding capture. Nonetheless, assuming that colonization happened around 2002, it seems that *F. heteroclitus* took no more than ca. 20 years to travel around 50 km from the Guadiana Delta, where it was first detected in 1976 (*Coelho, Gomes & Ré, 1976*)}, despite both areas being on a stretch of coast with a favourable benthic habitat (Fig. 4). This estimate indicates that if the Ebro colonization happened via natural dispersal, it must have happened 20 times faster than the natural colonization of the Ria Formosa. Given that *F. heteroclitus* has very low dispersal abilities (e.g., *Fritz, Meredith & Lotrich, 1975*; *Lotrich, 1975*), the natural colonization scenario for the Ebro seems unlikely.

Furthermore, if establishment in the Ebro Delta followed a natural range expansion one would expect to find several established populations between the Barbate marshes in the Guadalquivir, its southernmost location (*Gutiérrez-Estrada et al., 1998*), and the Ebro Delta. However, no invasive individuals were found at Mar Menor, a coastal lagoon with suitable conditions for successful establishment (*Pérez-Ruzafa et al., 2006*), located midway between the Guadalquivir and the Ebro Delta.

Finally, the Strait of Gibraltar would represent a barrier to dispersal and gene flow, limiting *F. heteroclitus* natural range expansion towards the Mediterranean Sea (*Doadrio, Carmona & Fernandez-Delgado, 2002*) because of the strong currents prevailing in the area and the absence of suitable habitats. Although uncertain for *F. heteroclitus*, this movement has already been documented for two other Iberian toothcarps, where restricted gene flow in this region led to speciation of the *Aphanius iberus* in the Mediterranean Iberian coast and *Aphanius baeticus* in the southwestern Atlantic Spanish coast (*Doadrio, Carmona & Fernandez-Delgado, 2002*).

### Ecological niche modelling

Conditions for the spread of *F. heteroclitus* are limited by the existence of benthic muddy saltmarsh environments compatible with the species' ecological requirements. These habitats are only found near major estuaries or lagoons areas along the Atlantic and western Mediterranean coastlines. The exception to this is along the North African coast (Fig. 4), where such environments are more continuous, although other aspects may be less favourable there, as suggested by the rapid decline in occurrence probability eastwards. The consequence of this benthic habitat constraint is to make along-shore colonisation unlikely in most areas, suggesting that any such sudden expansion would need to be aided by human release.

## CONCLUSIONS

*Fundulus heteroclitus* invasive Iberian populations revealed the presence of a single cyt *b* haplotype common to all individuals. This haplotype is predominant in the northern group of the native distribution, and although we cannot determine which exact location was at the origin of the introduced individuals, one can identify the northern end of range as the source of the introduction. The lack of genetic diversity is consistent with a strong founder effect at the origin of *F. heteroclitus* in Iberia. Although there is no direct evidence, we infer that the most likely vector was the aquarium trade, and that the Ebro Delta colonization results from a human-mediated secondary introduction isolated from the rest of Iberia. Considering the tolerance of the species to high salinity, its temperature range, and the significant amount of colonized area in the Iberian southern region, we predict that *F. heteroclitus* will most likely keep on expanding its invasive range until it faces unfavourable environmental conditions. However, natural colonisation in Europe will be strongly restricted by its restrictive requirement for suitable muddy, benthic habitats and human-mediated transfer is its most likely means of range expansion.

## ACKNOWLEDGEMENTS

Occurrence data for the AquaMaps modelling was provided through the AquaMaps website (http://www.aquampas.org), and the model simulation was run through the same site. Information on seabed habitats contained in this study has been derived from data that is made available under the European Marine Observation Data Network (EMODnet) Seabed Habitats project (http://www.emodnet-seabedhabitats.eu/), funded

by the European Commission's Directorate-General for Maritime Affairs and Fisheries (DG MARE). Authors are thankful to N. Franch and J.M. Queral (Parc Natural del Delta de l'Ebre, PNDE) for providing animals from the Ebro River Delta.

### Funding

The work was funded by funds to Rita Castilho from FCT strategic plan UID/Multi/04326/2016 granted to CCMAR. Regina Lopes da Cunha was supported by a post-doctoral fellowship (SFRH/BPD/109685/2015) from FCT (Fundação para a Ciência e Tecnologia, Portugal) and FSE (Fundo Social Europeu). North American sample collection was supported by NSF grant OCE-0221879 to David D. Duvernell. Authors are thankful to the PNDE for providing new fish samples. The funders had no role in study design, data collection and analysis, decision to publish, or preparation of the manuscript.

### Grant Disclosures

The following grant information was disclosed by the authors:
UID/Multi/04326/2016.
FCT (Fundação para a Ciência e Tecnologia, Portugal): SFRH/BPD/109685/2015.
FSE (Fundo Social Europeu).
NSF: OCE-0221879.

### Competing Interests

Rita Castilho is an Academic Editor for PeerJ.

### Author Contributions

- Teófilo Morim performed the experiments, analyzed the data, prepared figures and/or tables, authored or reviewed drafts of the paper, approved the final draft.
- Grant R. Bigg analyzed the data, authored or reviewed drafts of the paper, approved the final draft.
- Pedro M. Madeira performed the experiments, authored or reviewed drafts of the paper, approved the final draft.
- Jorge Palma and Enric Gisbert authored or reviewed drafts of the paper, approved the final draft, provided critical samples.
- David D. Duvernell authored or reviewed drafts of the paper, approved the final draft, provided critical DNA aliquots.
- Regina L. Cunha conceived and designed the experiments, analyzed the data, approved the final draft.
- Rita Castilho conceived and designed the experiments, analyzed the data, contributed reagents/materials/analysis tools, prepared figures and/or tables, approved the final draft.

### Animal Ethics

The following information was supplied relating to ethical approvals (i.e., approving body and any reference numbers):

Samples from USA were in the form of DNA aliquots originated from samples caught for previously published study in 2008: (Duvernell D.D., Lindmeier J.B., Faust K.E., & Whitehead A. (2008) Relative influences of historical and contemporary forces shaping the distribution of genetic variation in the Atlantic killifish, *Fundulus heteroclitus*. Molecular Ecology, 17, 1344–1360. Guidelines posted by the American Society for Ichthyologists and Herpetologists were followed. In Iberia, *Fundulus heteroclitus* is an invasive species. The individuals used in this study from Ebro Delta originated from field activities of eradication carried out by the Natural Park, no permits were needed. Other samples did not required specific permissions.

### DNA Deposition

The following information was supplied regarding the deposition of DNA sequences:
GenBank MH809691–MH809938.

### Data Availability

No raw data/code was produced.

### Supplemental Information

Supplemental information for this article can be found online at http://dx.doi.org/10.7717/peerj.6155#supplemental-information.

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
