# Peer review of "Invasion genetics of the mummichog (Fundulus heteroclitus): recent anthropogenic introduction in Iberia"

_PeerJ, doi:10.7717/peerj.6155_

## Round 0.1 · original submission · Minor Revisions

I now have reports back from 3 referees, all of whom are enthusiastic about the submission and the value of the work. However, each also point out some issues that could be improved prior to publication, so I am returning a decision of minor revisions. I look forward to reading your revised manuscript once you are able to address these comments.

Reviewer 1 ·

Basic reporting

The paper is extremely well written, clear and easy to follow. It has the expected structure, and the sequence data is available at Genbank.
As a general comment, I think that some parts of the paper are excessively long, and with too many citations. This happens in the Introduction, which could be reduced to include much less information about generalities on biological invasions. The information about Fundulus could be more restricted to genetic studies, but it seems to have the aim of reviewing all the published papers on the species instead.
In the discussion there is a similar pattern. Some interesting ideas are discussed, but because it is difficult to reach a conclusion, I would reduce the length of the arguments.

Experimental design

Cyt B (even if it was a larger fragment) was not an ideal option for this analysis. The authors should justify such decision, but the results are valid anyway.
The inclusion of the ecological niche modelling is weird. I am not an expert, so I cannot correct this part, but even if I think that these results should stay in the paper, I have the impression that they should be more connected with the other results, at least in the discussion.

Validity of the findings

The results are robust, even they don't provide much information. No definitive conclusions can be reached, neither on the origin of the Iberian populations (both northern and southern locations of the native range could be the origin), nor in the relationship between the three invaded sites (all 3 have the same haplotype).
I like the ideas in the discussion, but again, I think that they overdiscussed.

Additional comments

I liked the paper. There is an effort in the sampling of the native range, although the results do not reflect it, most likely because cyt B has too low diversity.

Reviewer 2 ·

Basic reporting

The manuscript "Invasive genetics of the mummichog (Fundulus heteroclitus): recent anthropogenic introduction in Iberia" aims to assess the invasion history of F. heteroclitus in the southern Iberian Peninsula.
The manuscript is well written, well referenced and overall scientifically sound.

Experimental design

The research question is relevant, however the methods used are not the most up-to-date. The authors made use of only a fragment of the mtDNA, for only three locations. These caveats, however, have been addressed in the discussion.

The novelty of the paper resides in the Ecological Niche Modelling approach used. However, this is not explored to its full capacity. I would recommend the authors to change the introduction/results/discussion to increase the focus on ENM.

The methodology is thoroughly described, and easily replicable.

Validity of the findings

The findings of only one major haplotype in the invaded range seem solid, and congruent with a recent invasion, mediated by a small number of individuals. The conclusions that this was done via aquarium trade might be a bit speculative.

I would strongly recommend to increase focus in the ENM.

Additional comments

The manuscript "Invasive genetics of the mummichog (Fundulus heteroclitus): recent anthropogenic introduction in Iberia" aims to assess the invasion history of F. heteroclitus in the southern Iberian Peninsula. The manuscript will make a contribution to the field of invasion biology, and I therefore recommend it to publication pending minor changes.

Minor comments:

line 51: there are reports of over
line 51: of which, 237
line 53: they are recognized as one of
line 75: remove extra paragraph
line 79: remove the between reconstruct and invasive
line 83: remove "all fulfilled for Fundulus heteroclitus" as the species is only introduced in the paragraph below
line 93: Fundulus in full at the beginning of the sentence
line 95: the presence of
line 106, 112, 116: add comma after )
lines 128-132: Although there is some interest in this study using only these aims, it is the ENM approach described in the methods that increase the novelty of this study. Please re-focus the introduction to reflect this angle.
line 156: remove the sequence of the primers, as they are described somewhere else
lines 182-219: This approach is not referred to in the introduction, and is not one of cited aims of the manuscript. Please include it, and the motivation behind itli
line 230: Figure 2 is referenced before Figure 1 - that should be changed
line 232: Figure 1a and 1b are not referenced in the text
lines 250-252: I recommend to expand this section, and elaborate on the findings of the ENM
line 280: This can be explored further. Either via ABC estimates of processes that would lead to maintaining such low genetic diversity. Is the species abundant in the invaded range or is it maintained at low levels? Are these long-term viable populations?
lines 310-311: This suggests that the population is actually not increasing in size
line 352: Which cannot really be discarded by the data used
line 362: Given the caveats of the data here presented, I would be weary to make this claim
lines 377-379: Which again suggests that these populations are not thriving in their invaded range
line 392: remove the
lines 405-413: Please expand this section, keeping in mind that the populations appear to have very limited ability to expand

Reviewer 3 ·

Basic reporting

The manuscript is very well-written. I only found a few minor language-related edits to make, which are indicated below. The level of background information provided was sufficient, and the overall structure of the paper was in accordance with the standards of PeerJ.

Minor edits:

-The title should read “Invasion genetics”, as opposed to “Invasive genetics”.

-Lines 40-44: this is a run-on sentence. It should be revised to read: “We suggest the most probable introduction vector is associated with the aquarium trade. We further discuss the hypothesis of a second human-mediated introduction responsible for the establishment of individuals in the Ebro Delta.”

-lines 44-46 (“Although the species…”) this part can be removed from the abstract. Otherwise, make it clear that it refers to the ecological niche modeling performed (and is not speculation).

-lines 51-52: what does “western European margins” refer to? I assume you mean “East Atlantic coastal areas”. Please revise accordingly.

-line 62: remove “non-native”.

-lines 109-110: according to information detailed below (lines 126-127), and other previous papers, macrolepidotus is the Northern subspecies, and heteroclitus is the Southern subspecies.

Experimental design

The authors emphasize well the originality of their results, and provide sufficient detail on how their work differs from previous publications (lines 128-132). Compared to previous work in this system, they have considerably expanded the number of populations and samples, as well as the length of the mtDNA fragment that was sequenced. The research questions are well defined. Lastly, the methods are described with sufficient detail to allow replication.

Validity of the findings

The data is robust, and the analyses are statistically sound. The conclusions are also well stated. I appreciated that the authors included a paragraph on caveats (lines 264-274), acknowledging the limitations of using only mtDNA information.

Additional comments

no additional comments to make.

---

## Round 0.2 · accepted · Accept

I have now read through your rebuttal and your revised manuscript, and feel that you have addressed all the concerns to my satisfaction. Thank you for your thorough responses to the referee concerns, and for selecting PeerJ for your submission.

#